**communications** engineering

# Talkative battery: super-safe batteries with power-modulation based internal and external sensor data collection
Johannes Diers[1] & Hamzeh Beiranvand [1,2] ✉

Individual cell surveillance in lithium-ion battery systems has not yet been widely adopted in the industry due to the increased cost of the final product. Thereby, the limited number of temperature sensors in a battery pack endangers system safety. This paper proposes talkative power conversion for collecting the temperature data from the sensors of a large-format battery cell, utilizing the high-frequency signal induced by the power converter modulation. Such a battery is called talkative battery in this paper. The principle of load shift keying (LSK) is utilized to establish a communication link between the sensor and the power converter (e.g., a battery charger) via the wire line. The concept of talkative battery is theoretically analyzed and experimentally validated with two different lithium iron phosphate (LFP) battery cells. The findings indicate that low-cost individual cell thermal sensing is viable with minimal hardware requirements.

The lithium-ion battery (LIB) is the most prominent energy storage solution in contemporary electronics and energy storage systems, owing to its high energy and power density. The growing demand for electric vehicles (EVs) and the necessity for large energy storage for electric grids that are equipped with lithium-ion batteries (LIBs) has resulted in a substantial increase in research interest concerning the enhancement of LIBs in various aspects. To ensure optimal performance, sustainability, and safety, meticulous attention must be paid to battery operation[1,2]. Given the strong impact of internal physics on battery performance, specifically its safety, accurate internal measurements are a valuable source of information. In particular, core temperature, electrode potential, internal pressure, strain, or gas formation measurements can be processed to increase cell safety and performance during operation[3]. Of these, thermal measurement is of particular importance in monitoring and preventing thermal damage to cells, as well as in assessing the cells' overall health[4–6]. Unregulated temperature can cause safety risks and reduce capacity due to the accelerated irreversible intercalation of lithium-ion, the decomposition of the SEI layer, or the plating of lithium[7,8]. In the context of high-energy cells, the maintenance of temperature is of particular significance for human safety. Thermal misuse can result in a phenomenon known as thermal runaway, which, in turn, can lead to the emission of potentially flammable gases.

The implementation of high-energy cells with larger volume and reduced cell count is an increasingly prevalent development in the EV industry, which applies to both cylindrical and prismatic cells[9]. As illustrated in Fig. 1a, there are several advantages and challenges associated with the large cell format. With a lower number of individual cells, their dedicated surveillance becomes easier to realize due to the lower amount of additional data channels. In parallel, due to the lower surface-to-volume ratio and the higher diameter of the large cell format, the core temperature of the cells is more likely to be affected by overheating[10,11], making individual cell supervision even more reasonable. A BMS that is solely equipped with external cell measurement data constitutes a cost-effective and easily implementable solution. This assertion is valid within the context of temperature measurement of small battery cells, wherein the disparity between the surface temperature of the cell and the temperature of its core is less pronounced. As the dimensions of the battery cell are increased, the external temperature measurement is no longer capable of providing accurate and latency-free internal temperature information, making the external temperature measurement risky. Employing mathematical models to estimate the core temperature using the current profile and surface temperature is impeded by the model limitation, since not all physical phenomena in the cell can be accurately represented, and the overall accuracy will degrade over time due to drifts in the electrochemical properties[12]. This suggests that internal cell temperature measurement is beneficial for large, high-energy cells. Therefore, a communication pathway between the battery cell and the BMS must be established.

Figure 1b depicts a variety of technologies, which have been employed to facilitate the transmission of temperature data from within a battery cell to external systems. Examples of such technologies include the Controller Area Network (CAN)[13,14], Power Line Communication (PLC)[15,16], or Capacitive Coupling[17]. The transfer of data between battery cells and the BMS is typically regarded as constituting measurement data. However, it is

[1]Chair of Power Electronics, Kiel University, Kiel, Germany. [2]Kiel Nano, Surface and Interface Science (KiNSIS), Kiel University, Kiel, Germany.
✉e-mail: hab@tf.uni-kiel.de

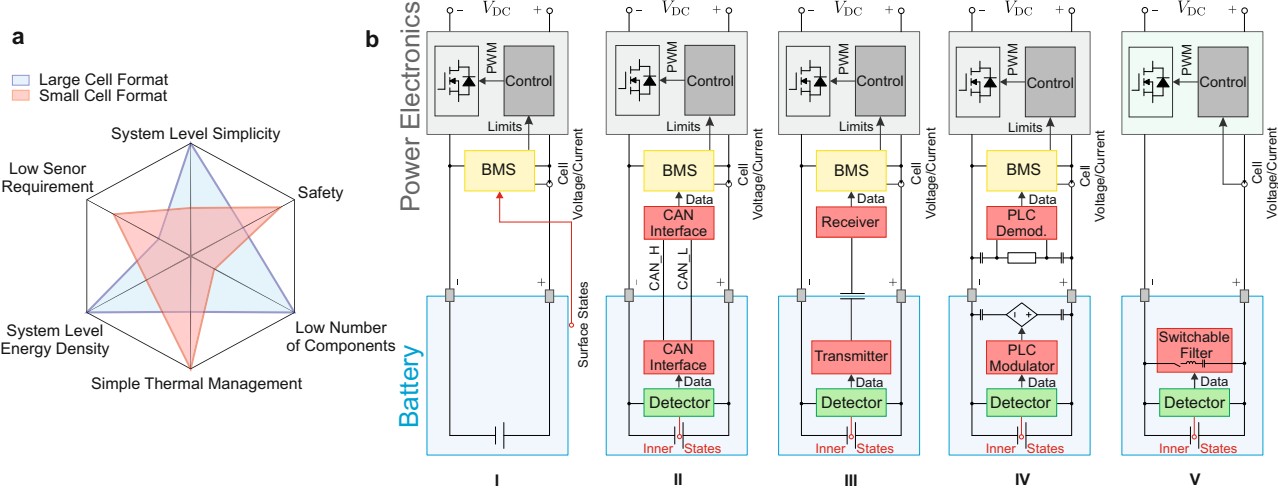

**Fig. 1 | Motivation for the development of the proposed data transmission strategy. a** Aspects of cell integration for large and small cell formats. **b** Different communication technologies in batteries. (**I** Simple Battery Management System (BMS) without communication to inside the cell, **II** Controller Area Network (CAN), **III** Capacitive coupling communication, **IV** Power Line Communication (PLC), and **V** the proposed communication strategy for internal sensing.

noteworthy that this data may also encompass control instructions, such as those employed for the purpose of balancing the battery voltages[18]. The implementation of such measures does not necessitate a high data rate. Serial communication performed with CAN provides efficient cabling, robustness, and automatic error correction as stated in ref. [19]. Another strategy is to use capacitive coupling through the cell case, using a metal film outside the cell, to establish a communication channel[17,18]. These two technologies require additional cables, which consume material and space, making their implementation disadvantageous. Therefore, the focus should be on technologies that do not use extra cables. The first option that comes to mind is radio, using standards such as Bluetooth or ZigBee. A salient disadvantage of these solutions is their reliance on an antenna, in conjunction with the absence of a metal housing that might block the electromagnetic waves[20,21]. An alternative method of avoiding additional wiring is to utilize the electrical connections of the cells that facilitate power transfer for data transfer, e.g., PLC. In PLC the data signal is superimposed on the power-bearing electrical quantities using capacitive or inductive coupling and a carrier. The selection of the frequency is of paramount importance in ensuring that interference between carrier and energy transfer-related quantities is avoided[13,22]. Each participant is required to have a modem with transceiver electronics. PLC for battery cell data transmission has been implemented in ref. [15] with a capacitive coupled PLC modem in every cell and at the BMS. Using PLC additionally has the drawback that it is realized by applying the carrier on the battery poles, which hold a low impedance. Hence, PLC should be turned off when the battery is resting to avoid unnecessary energy waste from the carrier.

Additional hardware reduction can be achieved through the use of power-modulation-based communication, otherwise referred to as talkative power conversion[20,23,24]. This strategy involves the utilization of converter-induced ripple as a carrier for data transmission. Various modulation schemes, such as frequency-shift-keying or phase-shift-keying, can be applied[23]. The integration of such technology could facilitate a transmission pathway from the converter to the battery cell. Nevertheless, the principal objective is to facilitate data transmission from the battery cell to the attached electronics. This is not possible when the current talkative power concept is employed due to the nature of battery cells delivering only direct current (DC).

In accordance with the aforementioned points, this paper puts forth a power-modulation-based concept for data transmission from the battery cell to the BMS. The data transmission is achieved by generating the carrier at the power converter side using the converter-induced current and voltage ripple and manipulating the ripple at the cell side by altering its impedance using a miniature switchable LC resonant circuit, the switchable filter. It is connected in parallel to the battery terminals so as not to affect the battery's electrical performance, and it is small enough to be incorporated into typical large-format prismatic cells. This system constitutes a power-modulation-based data transmission system, as it facilitates data transmission from the battery to the carrier source, which is the power converter, while the classical talkative power conversion concept identifies the power converter as the data transmitter. The theoretical foundation has been established, and a design procedure presented. The subsequent validation of the proposed concept is accompanied by a detailed exposition of its implementation in two experimental case studies, one for internal temperature sensing and one for external temperature sensing. The external sensing is validated using a 100 Ah battery cell (CA100AHA, manufactured by CALB Co. Ltd) and the internal sensing using a customized manufactured battery cell. The results demonstrate the successful transmission of data from the battery to the converter with a high signal-to-noise ratio, even for sufficiently long connecting cables.

## Results

### Battery temperature sensing

The complex electrochemical dynamics of a LIB cell are contingent on multiple physical quantities, with temperature playing an intense role. The phenomenon of battery heating can be attributed primarily to irreversible Joule heating, which accounts for over 70% of the total heating effects observed. This is supplemented by exothermic reactions resulting from electrolyte breakdown during periods of high charging state[25,26]. The heat generation in the electrodes that exists due to the absorption and release of lithium ions is called entropic heat and is a reversible process that is contributing negligibly to the cell heating[26]. The BMS can regulate the battery temperature by modifying the current, which is directly related to the internal battery heat generation. This enables the BMS to maintain the battery's temperature within the desired range. The optimal operating temperature range for LIB is defined by several reactions that can be triggered at the electrodes or the electrolyte. Below 15 °C or above 35 °C, the performance of the LIB tends to reduce, which can be seen by a facilitated degradation process of the capacity, increase of electric impedance, or reduction of efficiency[8,27,28].

Consequently, precise temperature measurement is essential for the BMS in order to ensure the effective and sustained implementation of a LIB. Furthermore, the measurement of temperature is of particular significance

in ensuring the safety of the battery. Thermal breakdowns can be detected or even prevented by intervening immediately in the battery's operation[29,30]. Measurement of the surface temperature of the battery cell is the simplest approach to collect thermal information from the cell because temperature sensors can be easily mounted on the cell case. The operation of discrete electric sensors, such as thermocouples or resistive temperature detectors with positive (PTC) or negative temperature coefficient (NTC), is due to their high availability and well-known behavior commonly applied[29,31,32]. A range of alternative temperature measurement technologies has been developed for use in the research field, including optical fiber-based temperature sensors that utilize Bragg grating[5,33]. However, the need for an optical transducer means that implementing them with a BMS for use in the field is not ideal. A new approach in sensor production is to use temperature-sensitive inks, which carry, for example, carbon-based nanomaterials or conductive polymers, and print them on a desired substrate[34–36]. Such sensor technologies allow not only a more flexible application of sensors on the battery surface, but also an easy implementation inside the cell case at the origin of the generated heat, where the highest temperature can be expected. The principal benefit of internal temperature measurement is that it is not impaired by a long time delay, as is the case for external measurement. Furthermore, it provides the exact value of the more relevant core temperature. The difference is demonstrated in refs. [12,37], in which implanted resistive temperature detectors were compared with external detectors. This leads to the next level of development, which is the application of the sensor on the battery components, such as the electrodes or the separator. In ref. [38], a sensor for dissolved manganese ions has been printed on the separator of the battery. The placement of a temperature sensor on the separator in a similar manner would facilitate the acquisition of the most immediate temperature value of the electrolyte and the electrodes. The capacity of the BMS to detect the core temperature and other critical internal battery parameters constitutes the foundation for enhancing the safety and durability of the battery. This, in turn, is a key motivation for the development of this sensor data transmission technology.

## Load-shift-keying

As mentioned before, in classical power-modulation-based talkative power, the generator of the induced voltage or current ripple is also the transmitter of the data[23]. This is not directly applicable to the proposed problem, since battery cells naturally provide DC power and no additional ripple. This prompts the question of whether there exists a method for employing power modulation in which data is transmitted from the load side, while the receiver generates the ripple, which is then utilized as a carrier for transmission. One potential solution to this problem is to vary the load impedance. This technique is most commonly referred to as load-shift-keying (LSK)[39–41].

LSK was considered in 1995 by Tang et al. for implantable telemetry systems[39]. Such devices are used to detect heart rate, applying electrocardiograms, electroencephalographs, temperature measurement, and to record other valuable medical information[39]. As these implantable devices depend on an external wireless power source, they are connected by inductive coupling to a power supply placed outside the patient's body. The frequency of the applied alternating current is 8.75 MHz. As data from the implanted device needs to be read out, the inductive link can also be used for data transmission[39]. In order to achieve this objective, Tang et al. employ the LSK technique in conjunction with a circuit configuration modulator. This modulator effects a modification to the rectifier circuit within the telemetry device, thereby rendering it a full-wave rectifier during the transmission of a zero and a voltage clamp during the transmission of a one[39].

This technology has been improved by Karimi et al. with the implementation of band-pass filters[41]. Their primary function is to differentiate between the main carrier, which is employed for the transmission of electrical power, and the auxiliary carrier, which is utilized for data transmission. The modulation of the bits is achieved through the activation and deactivation of the load positioned behind the communication band-pass filter, which corresponds to the binary digits one and zero[41]. Also Yilmaz et al.

implemented a derivative of the frequency-sensitive LSK modulator[40]. The resonant frequency of the load, which contains a parallel resonant circuit with a coil and a capacitor, is detuned by the modulator. The addition of a second capacitor in parallel with the primary capacitor of the resonant circuit results in a slight diminishment of the resonant frequency, thereby modifying the impedance characteristics. Whilst the PWM is being applied at the original resonant frequency, it has been demonstrated that there is an increase in impedance once the capacitor has been connected because the circuit has been detuned[40]. The issue of communication between the interior of LIB cells and external electronics, particularly the BMS, has not yet been addressed using LSK technology.

## Proposed communication principle

The concept of LSK is applied to the battery, which has two poles that maintain a certain frequency and state of charge (SOC) dependent impedance. Figure 2 illustrates the fundamental idea, in which a PWM-controlled buck converter that controls the battery energy flow causes a current ripple $i_c(t)$ at a separate frequency band. The talkative battery shifts its impedance, which is detected on the outside by an alteration of the current amplitude and phase. To change the impedance of the LIB cell, an LC resonant filter is connected in parallel with the cell. This filter can be switched off using an electrically controlled switch, such as a MOSFET.

## Battery impedance manipulation

The impedance of the battery cell (CA100AHA) is initially determined by electrochemical impedance spectroscopy (EIS), which facilitates the investigation of the potential impact of the switchable filter on the cell impedance and the simulation of the proposed LSK transmission system. The investigated battery cell reaches a minimum impedance of 1 mΩ at 300–350 Hz and a low SOC, but rises with increasing frequency due to internal inductive effects. The overall low impedance in Fig. 3a indicates that the switchable filter must have an impedance at resonance that is small enough to affect the total impedance of the parallel arrangement. It is preferable for the operating frequency to be elevated, as the battery impedance increases with frequency.

The utilization of an equivalent circuit model (ECM) that incorporates a SOC-dependent voltage source and an impedance model realized with resistors, inductors, and capacitors facilitates the generation of an electrically precise digital replica of the cell. Batteries exhibit capacitive behavior at low frequencies due to the parallel orientation of the current collectors and the high permittivity of the electrolyte and electrode materials used. Up to a certain frequency, they are generally modeled using capacitive elements, like a parallel RC element in series to the remaining ECM. The dynamics of Li-ion transport by diffusion, polarization effects, or electrochemical reactions are often implemented by additional modeling elements in the circuit[42,43]. For the cell studied, the frequency at which the impedance begins to become inductive is in the range of 190–240 Hz. Then the impedance increases due to the inductive properties of the cell geometry. Often, a constant phase can be seen, resulting from the proportional rise of the real and imaginary components[44], which is also the case for the cell studied. Its phase remains just below 80° up to 500 kHz and the absolute value increases linearly with frequency. This behavior can be modeled using parallel RL elements in series to the open circuit voltage, which has been employed previously by Ferraz et al., who additionally utilized serial RC parallel elements to facilitate low-frequency applications[44]. For the proposed problem, a model with three RL parallel elements and a single resistor and inductor serial connection has been chosen, as shown in Fig. 3b. The model is fitted to the EIS measurement result of the CA100AHA LFP cell using particle swarm optimization. The fit function is defined as the sum of the squared absolute values of the impedance deviation, which is then normalized to the absolute value of the actual impedance. It is applied in the range 1.5–237 kHz over 36 data points.

To study the electrical effect of the switchable filter, its admittance with the battery impedance in parallel, defined as $\underline{Y}_{sc} = \frac{1}{\underline{Z}_{sc}}$, is examined. In this simulation, filter parameters are adjusted to $L_f = 0.8\,\mu H$ and $C_f = 600\,nF$. The parasitic resistive components in the switch, inductor, and capacitor of the

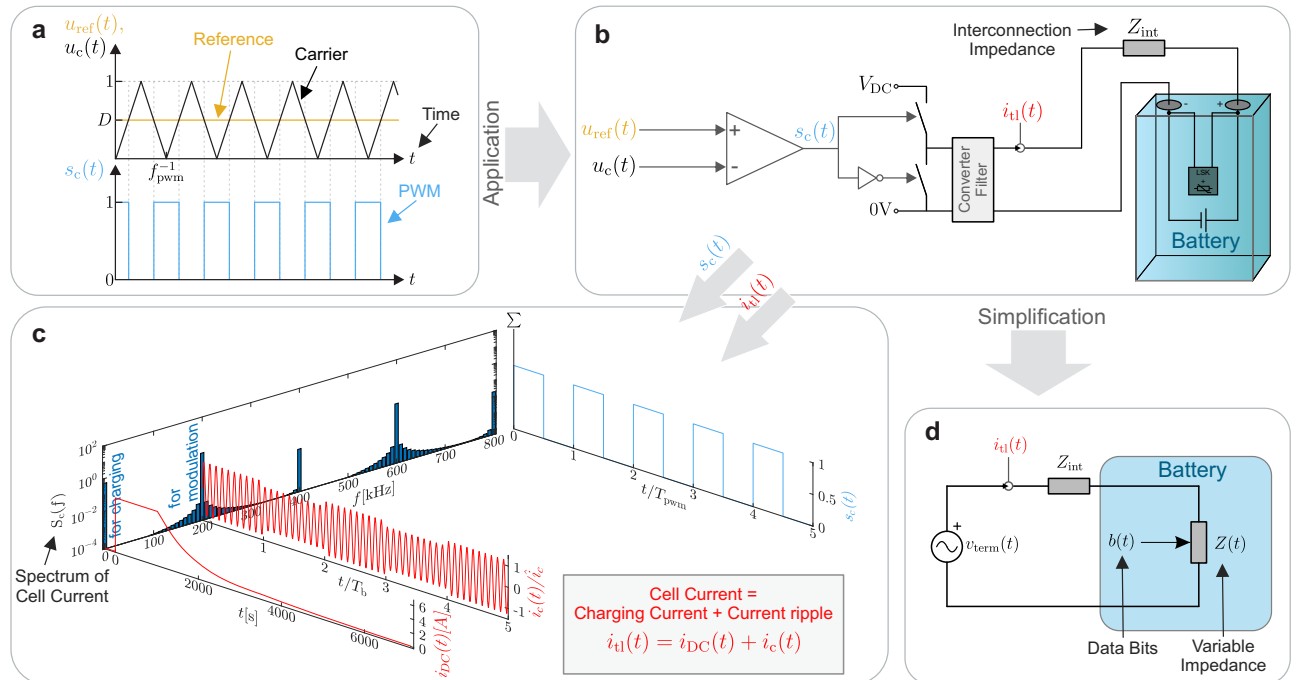

**Fig. 2 | The LSK concept to transfer temperature data from the cell inside to the outside. a** Signal plot of the PWM generation $s_c(t)$ with its fundamental frequency $f_{pwm}$. **b** Schematic of the power modulation based LSK concept with interconnected modulation circuitry, half bridge converter and battery. **c** The PWM spectrum, denoted by $S_c(f)$, generates frequency-related components of the current. These components include low-frequency components that facilitate the charging process,

$i_{DC}(t)$, and current ripple resulting from the first harmonic of the PWM, $i_c(t)$. The letter can be employed for LSK. **d** Simplified circuit model for demonstration of the LSK concept with a variable battery impedance $Z(t)$ and an intermediate impedance $Z_{int}$ serially connected to an AC voltage source. The impedance shift of the battery can be detected from the outside by measuring the change in the current ripple.

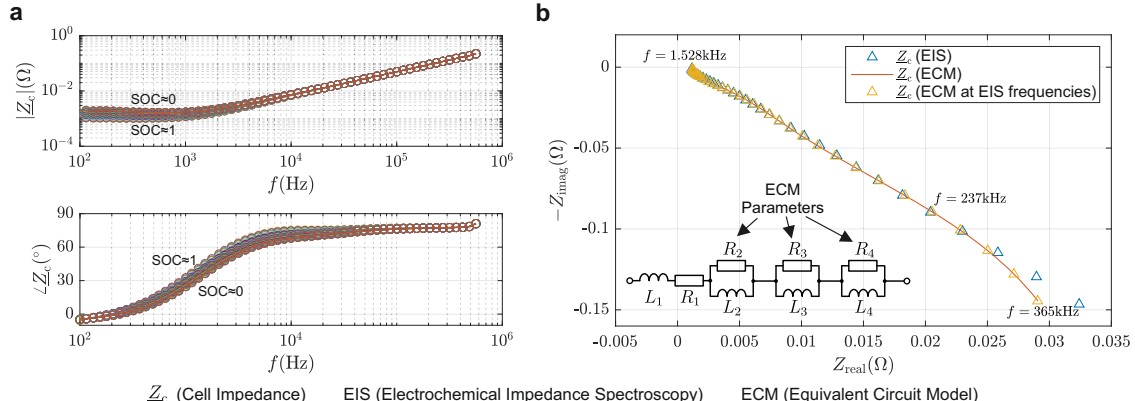

**Fig. 3 | Impedance analysis of the examined battery. a** The bode plot of the battery cell impedance $\underline{Z}_c$ measured with Electrochemical Impedance Spectroscopy (EIS) shows a little dependence on the State of Charge (SOC), but the frequency $f$

dependence is high, as it ranges around 1 mΩ between 100 Hz and 1 kHz and rises for higher frequencies. **b** Impedance fit of the Equivalent Circuit Model (ECM) to the EIS measurements and circuit of the ECM used for simulation.

switchable filter are modeled by implementing the resistor $R_f$ in series with the other switchable filter elements in the simulation. An optimistic value of $R_f = 50$ mΩ is assumed. $\underline{Y}_{sc}$ shows that at the resonant frequency of the filter, which is 230 kHz, the admittance delineates a circle on the Nyquist diagram. As the admittance of only the LCR resonant filter describes a circle with the diameter of $\frac{1}{R_f}$ starting at the origin of the Nyquist plot, we can see that this circle is superimposed on the admittance curve of the battery cell, as shown in Fig. 4a. The frequency $f_{opt}$ with the maximum admittance shift is very close to the resonant frequency of the switchable filter. This bit-controlled impedance shift is detected on the converter side. In order to do so, the admittance of the connection cable between cell and power electronics must

be taken into consideration, as well as the properties of the attached power electronics. It is common practice for power electronics to be equipped with an output filter. Therefore, the complete circuitry, depicted in Fig. 4b, will be analyzed subsequently.

## Circuit analysis

The power electronics are represented by a bidirectional buck converter, which contains a half-bridge and an LC-output filter. The half-bridge on the left-hand side of Fig. 4b is supplied with a constant voltage $V_{dc}$. The converter output filter is equipped with an inductor $L_F$ and a capacitor $C_F$. The connection cable between the converter and the battery cell is modeled with an inductor $L_{tl}$ and a resistor $R_{tl}$. The talkative battery holds the

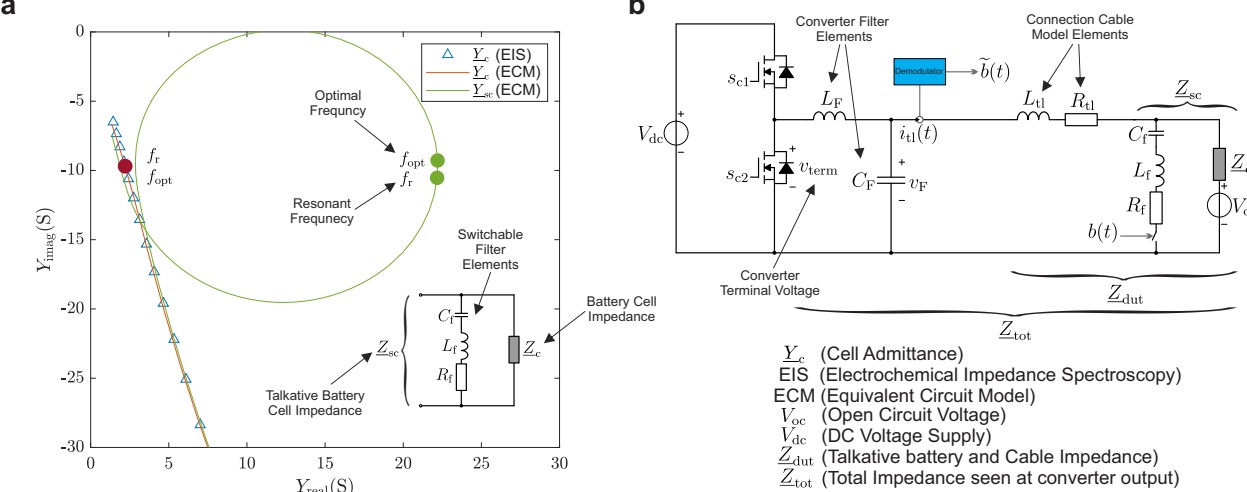

**Fig. 4 | Implementation of the switchable filter. a** Nyquist plot with the admittance of the battery $\underline{Y}_c$ and the parallel connection of filter and battery $\underline{Y}_{sc}$ with resonant frequncy of $f_r$ = 230.00 kHz and a maximum admittance shift at $f_{opt}$ = 229.6 kHz. **b** Circuit model of the transmission system with converter, converter filter with $L_F$

and $C_F$, the connection cable model with $L_{tl}$ and $R_{tl}$, and battery cell with switchable filter, represented by $\underline{Z}_{sc}$. $\underline{Z}_{dut}$ represents thereby the impedance of the talkative battery with the transmission cable. $\underline{Z}_{tot}$ corresponds to the whole circuit impedance connected to the converter leg.

communication module with the filter with $C_f$, $L_f$ and $R_f$ in series to a switch, which is implemented in the experiments using a MOSFET. The impedance of the talkative battery, which is the parallel connection of the switchable filter impedance $\underline{Z}_f$ and the battery impedance $\underline{Z}_c$, is denoted as $\underline{Z}_{sc}$. The effect on the battery current ripple $\hat{\underline{i}}_c$, which will be detected for demodulation, is expressed by the transfer function $\underline{H}_{CLSK}$. The impedance $\underline{Z}_{tot}$, which represents the impedance of the complete circuitry that is connected to the output side of the converter MOSFET-leg, will be considered as constant since it is dominated by $L_F$. The transfer function analysis in formula (3) shows the nonlinear effect of the switchable filter on the converter output current. It is embedded in a fraction where the converter filter capacitor and the parasitic cable elements have a strong impact. The analysis of the embedded admittance $\underline{Z}_{dut}$ has shown that the frequency $f_{opt}$, at which a high ripple change in the current is to be achieved, can deviate from the resonant frequency of the switchable filter due to the connection cable model. Therefore, in the experimental section, the manual adjustment of $f_{pwm}$ has been performed to maximize the shift of $\hat{\underline{i}}_c$ induced by the impedance change. The symbol $b \in 1, 0$ represents the current bit to be transmitted.

$$\underline{Z}_f = \frac{1}{j\omega C_f} + j\omega L_f + R_f \tag{1}$$

$$\underline{Z}_{tot} = j\omega L_F + \cfrac{1}{j\omega C_F + \cfrac{1}{j\omega L_{tl} + R_{tl} + \cfrac{1}{\frac{1}{\underline{Z}_c} + \frac{1}{\underline{Z}_f} \cdot b}}} \tag{2}$$

$$\underline{H}_{CLSK} = \frac{\hat{\underline{i}}_{tl}}{\hat{\underline{v}}_{term}} = \frac{\frac{\frac{1}{j\omega L_{tl}+R_{tl}+\frac{1}{\frac{1}{\underline{Z}_c}+\frac{1}{\underline{Z}_f}\cdot b}}}{j\omega C_F + \frac{1}{j\omega L_{tl}+R_{tl}+\frac{1}{\frac{1}{\underline{Z}_c}+\frac{1}{\underline{Z}_f}\cdot b}}}}{\underline{Z}_{tot}} = \frac{\frac{\frac{1}{\underline{Z}_{dut}}}{j\omega C_F+\frac{1}{\underline{Z}_{dut}}}}{\underline{Z}_{tot}} = \frac{\frac{1}{j\omega C_F \underline{Z}_{dut}+1}}{\underline{Z}_{tot}} \tag{3}$$

To obtain a comprehensive understanding of the necessary set of switchable filter capacitor value and excitation frequency that causes a maximum current ripple change, the impact of the filter is defined as the normalized change of the transfer function, denoted by $\underline{H}_{CLSK}$. This transfer function exists in two distinct states. The circuitry gets represented by $\underline{H}_{CLSK.on}$ when the switch of the switchable filter is conducting ($b=1$) and by $\underline{H}_{CLSK.off}$ when the switch is blocking ($b=0$). Accordingly, the deviation is

represented by:

$$\Delta H_{CLSK} = \left| \frac{\underline{H}_{CLSK.on} - \underline{H}_{CLSK.off}}{\underline{H}_{CLSK.off}} \right| \tag{4}$$

$\Delta H_{CLSK}$ is calculated by implementing the ECM of the CA100AHA cell for $\underline{Z}_f$, the cable impedance model of a 300 mm cable, which is later used in the experiments and parameterized with $L_{tl}$ = 212 nH and $R_{tl}$ = 5 mΩ, the desired switchable filter inductance $L_f$ = 0.8 μH, measured resistance $R_f$ = 80 mΩ, converter filter capacitance $C_F$ = 3.3 μF and inductance $L_F$ = 7 μF. In the subsequent analysis, the switchable filter capacitance and converter switching frequency have been applied as a variable to see the effect on the filter influence $\Delta H_{CLSK}$.

The green-yellow sectors in Fig. 5a with high $H_{CLSK}$ variation can be separated into three parts. By applying a switchable filter capacitor with a capacitance of over $C_f$ = 1 μF the impact $\Delta H_{CLSK}$ has a maximum at an almost constant excitation frequency of $f_{sw}$ = 175 kHz, which indicates, that the resonance occurs between the converter filter capacitor and $\underline{Z}_{dut}$, which is almost inductive. The activation of the filter changes $\underline{Z}_{dut}$ and hence the resonance frequency. The application of the proposed method in region A is disadvantageous in that, by configuring different resonant frequencies in several cells, it is impossible to distinguish between them by applying different excitation frequencies. In region C, the maximum of $\Delta H_{CLSK}$ is located at different frequencies $f_{sw}$ for various values of $C_f$, which shows that the resonant frequency of the switchable filter mainly defines the optimal frequency. An increase of the cable inductance $L_{tl}$ results in a decrease of $\Delta H_{CLSK}$ in this region, since the cable impedance dominates over $\underline{Z}_{sc}$. The transition region B is where both resonant effects cross, which gives the highest $\Delta H_{CLSK}$ but also a low dependence on $C_f$. By implementing a higher converter filter capacitance of $C_F$ = 5 μF, the resonance of the converter filter capacitor with $Z_{dut}$ appears at a lower frequency, which shifts the transition region to the left upper corner, as illustrated in Fig. 5b. This enables an operation in region C at lower frequencies. The operation of LSK regarding the proposed issue can be applied in all three regions, especially when a long cable with high inductance is applied, region A is expected to still provide a high $\Delta H_{CLSK}$. The frequency selectivity feature, however, is only applicable at the maxima of regions B and C.

The circuit depicted in Fig. 4b has been simulated using MATLAB Simulink and PLECS in order to demonstrate the feasibility of the proposed method. The model of the used battery CA100AHA has been implemented

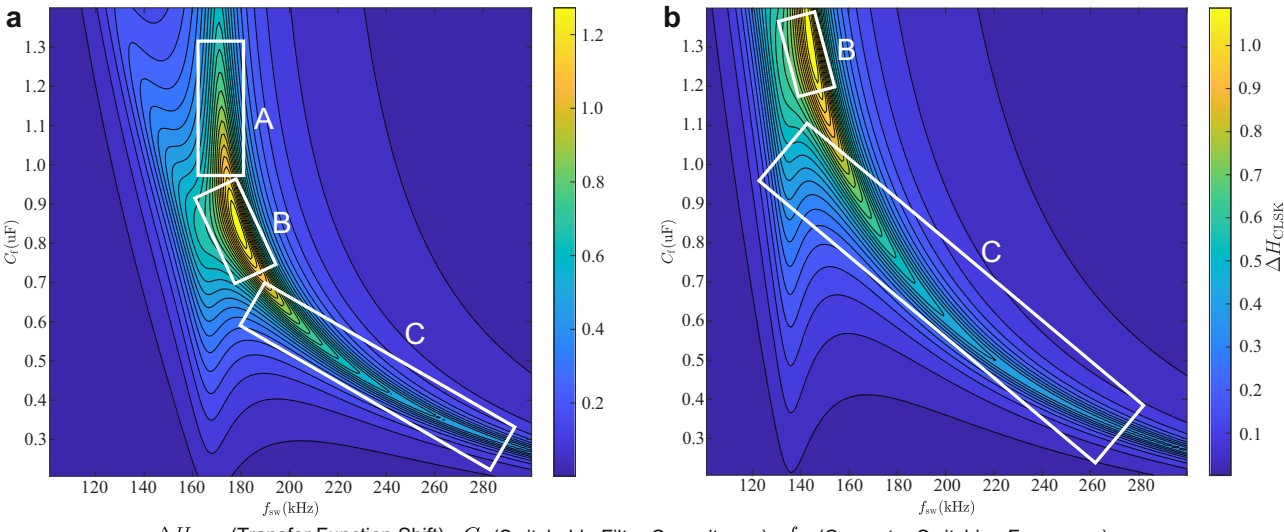

**Fig. 5 | $\Delta H_{\mathrm{CLSK}}$ as a variable dependent on the switching frequency $f_{\mathrm{sw}}$ and switchable filter capacitance $C_{\mathrm{f}}$ with a converter filter capacitance of: a $C_{\mathrm{F}}$ = 3.3 μF and b $C_{\mathrm{F}}$ = 5 μF.**

**Fig. 6 | Simulation of the proposed system.** The plot of the transferred data stream $b(t)$, the battery voltage $v_{\mathrm{bat}}(t)$, the battery current $i_{\mathrm{tl}}$, and the signal generated by synchronous demodulation of the cable current $u_{\mathrm{dem}}$ are shown.

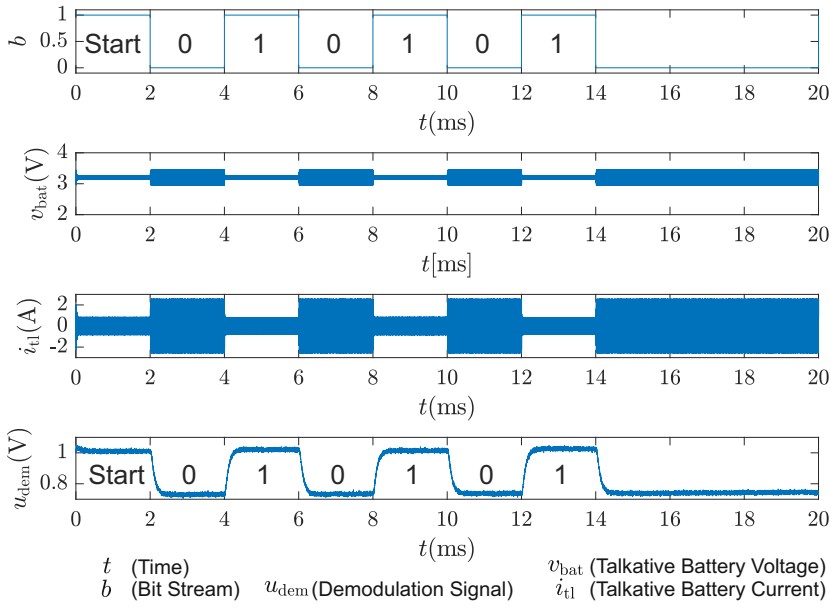

into the simulation. The power cable is modeled with $L_{\mathrm{tl}}$ = 70 nH and $R_{\mathrm{tl}}$ = 2 mΩ, which matches the impedance of a 70 mm double cable, also used in the experiments. The converter filter is set to $C_{\mathrm{F}}$ = 3.3 μF and $L_{\mathrm{F}}$ = 7 μH. When 227 kHz is used as the switching frequency, the data signal, which has a bit length of 2 ms, can be seen in the change of the current ripple. The current is sensed using a current-controlled voltage source with a transfer ratio of 0.59 V A$^{-1}$. The coherent demodulation applied on the current sensor output voltage using a phase-sensitive detector and an RC lowpass filter of second order with the parameters $\tau_1 = \tau_2 = 47$ μs shows, that the achievable voltage signal $u_{\mathrm{dem}}$ reaches a shift of 180 mV, as shown in Fig. 6. The phase shift of the phase-sensitive-detector (PSD) PWM is adjusted manually to $\phi_{\mathrm{dem}} = 280°$ to reach the highest voltage level difference on $u_{\mathrm{dem}}$.

It has been observed that the current ripple during the data period, in which the filter is deactivated, exceeds the levels observed during periods in which the filter is activated. This phenomenon is likely attributable to the

resonance between the converter output filter capacitor and the cable and battery impedance, which is detuned by the switchable filter.

**Setup**

The circuit, which contains the functionality of the battery temperature measurement and, in parallel, realizes the LSK through the switchable filter, has been realized on a small PCB. The NTC is read out using a voltage divider with a 10 kΩ resistor, and the conversion into a digital bitstream is performed with an ATmega328P micro-controller. The switchable filter is designed with the parameters $C_{\mathrm{f}}$ = 600 nF and $L_{\mathrm{f}} \approx$ 0.8 μH. The converter filter instead is designed with $C_{\mathrm{F}}$ = 3.3 μF and $L_{\mathrm{F}}$ = 7 μH. The battery CA100AHA was first attached in parallel with the switchable filter to test the proposed topology using the battery with very low impedance. In the experiments, two different copper double cables of 70 mm and 300 mm length have been tested. The cable has been attached to the battery on one side and to the current sensor and converter filter board on the other. The

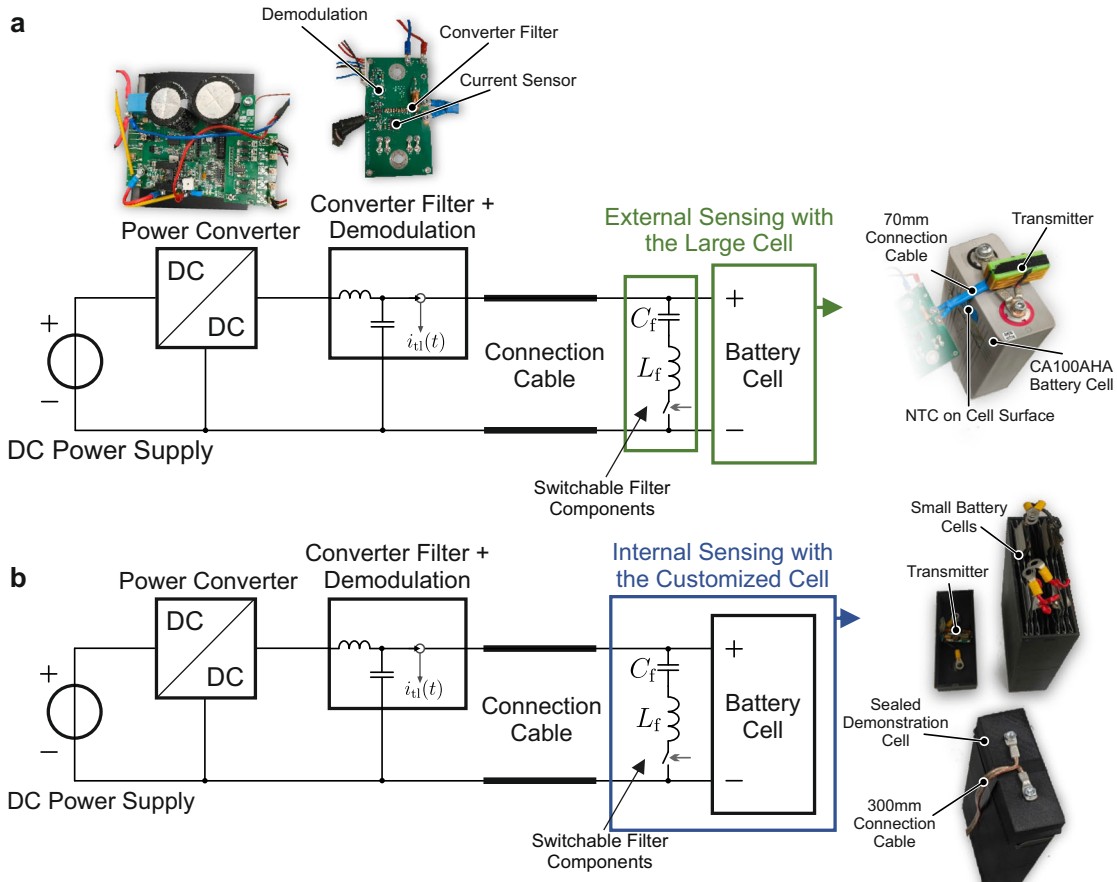

**Fig. 7 | Description of the experimental setup.** The components are the power converter, the converter filter, the current sensor with demodulation circuitry and the transmitter with **a** the large CA100AHA battery cell with external sensing, or **b** the customized demonstration cell with internal sensing.

coherent demodulation is located separately on the latter. Demodulation is realized with a phase-sensitive detector. The TMS320F28379D micro-controller by Texas Instruments has been utilized to facilitate the pulse-width modulation (PWM) for the converter and demodulation processes. Concurrently, the micro-controller undertakes the task of bit detection from the demodulation output and provides an indication signal $u_{\mathrm{det}}(t)$ to signalize when a start-bit or one is detected. In addition, the transmitter should be integrated into a battery cell. Given the risks associated with opening the proposed battery cell, a new battery pack has been assembled. 6 LFP cells with a capacity of 8 Ah have been connected in parallel inside a housing, which has been created using a 3D printer. The miniaturized transmitter was then implanted into the housing and subsequently connected to the cells. Both setups, with external and internal sensing, are illustrated in Fig. 7.

### Experiments

Two groups of experiments were conducted: one for a commercial CA100AHA and one for a customized cell for external and internal sensing, respectively. The voltage $V_{\mathrm{dc}}$ is set to $\approx 7\,\mathrm{V}$ while the duty cycle of the converter is always 0.5, which keeps a constant voltage and no current on the battery side. In both case studies, a bit period of 2 ms has been used, while every message contains 10 bit and the time interval of the messages being transmitted is 43 ms. Hence, the data rate is 232 bit s$^{-1}$. and the effective bandwidth, which incorporates the transmission and reception time, is 153 bit s$^{-1}$. The experimental results are detailed in the following subsections.

### External sensing

The first experiment with the miniaturized transmitter aims to prove the proposed concept with the LFP cell CA100AHA. In this setup, the 70 mm

connection double cable is connected, and the transmission unit with the temperature sensor and the switchable filter is attached to the battery poles outside the cell. After adjusting the switching frequency, the change of the demodulation output voltage of 180 mV could be seen. The switching frequency is set to 262 kHz, where the highest impact on the demodulation signal could be observed. This experiment confirms the simulation result that the activation of the switchable filter does not necessarily result in a current amplitude increase, but can also result in a decrease of the current amplitude $\hat{\underline{i}}_{c}$. By increasing the input voltage to $V_{\mathrm{dc}} = 13.4\,\mathrm{V}$ an input current of the converter of 5.16 A could be generated. With the duty cycle of 0.5, the average battery current reaches $i_{\mathrm{DC}} = 10.32\,\mathrm{A}$ and data transmission could still be established. Figure 8a, b show the demodulation output, the micro-controller's indication for message start and bit detection, and the modulated output of the current sensor for both examined cases.

### Internal sensing

A second experiment is carried out with the same transmitter inside the demonstration battery cell. The switching frequency is set to $f_{\mathrm{pwm}} = 250\,\mathrm{kHz}$. Transmission is possible for both cable lengths, even when the battery is simultaneously charged at a constant current of 12 A. Figure 8c, d show the relevant waveforms with the customized cell and the 70 mm connection cable. For the case without load current, the demodulation signal reached even a voltage difference of 555 mV. Also here, the application of a load current does not hinder the data transmission, as depicted in Fig. 8d.

### Quantitative comparison

Since the performance of the impedance shift detection is to be evaluated in conjunction with the demodulation strategy, the demodulated signal must

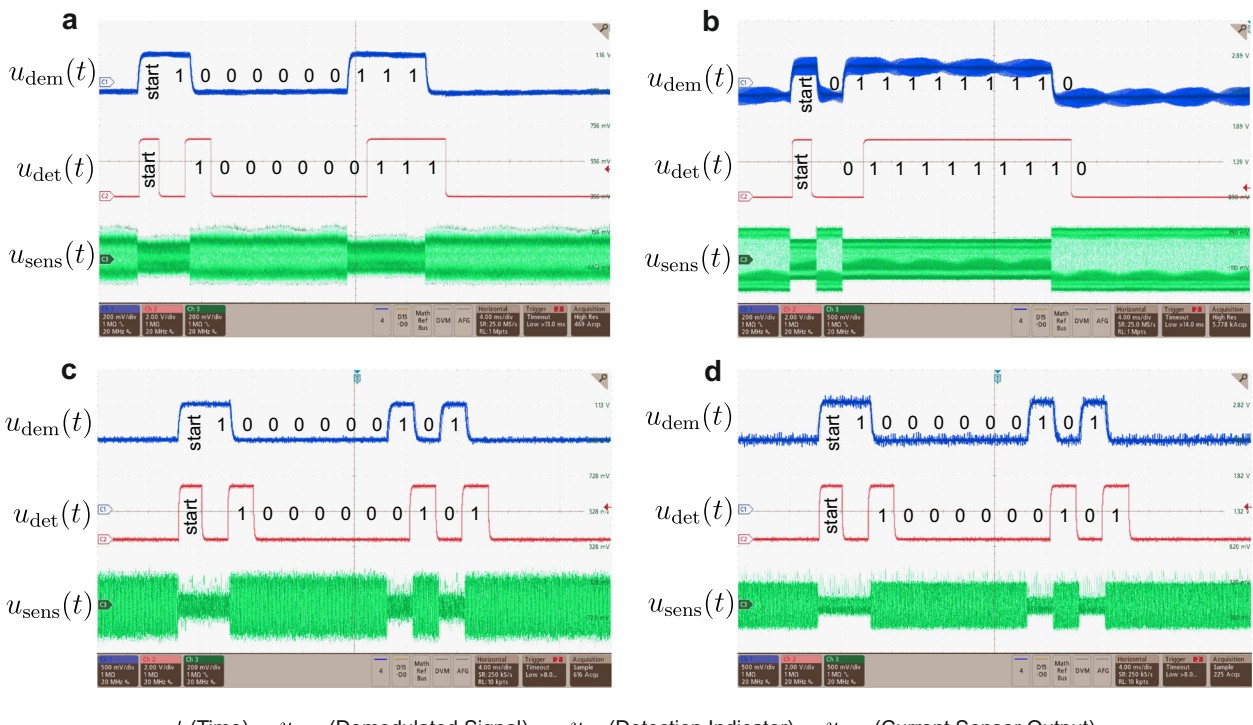

$t$ (Time)   $u_{\text{dem}}$ (Demodulated Signal)   $u_{\text{det}}$ (Detection Indicator)   $u_{\text{sens}}$ (Current Sensor Output)

**Fig. 8 | Demodulator output $u_{\text{dem}}$, detection indicator $u_{\text{det}}$, and current sensor output $u_{\text{sens}}$ during data transmission via LSK with a 70 mm double connection cable. a** with commercial CA100AHA battery cell and zero average battery current ($i_{\text{DC}} = 0$ A), **b** with CA100AHA battery cell and average charging current of $i_{\text{DC}} = 10.32$ A **c** with the customized cell and zero average battery current ($i_{\text{DC}} = 0$ A), **d** with the customized cell and average charging current of $i_{\text{DC}} = 10.30$ A.

**Table 1 | SNR, SER, and BER (based on equation (5)) of the demodulation signal for the simulation and the experiments with the embedded miniaturized transmitter for internal and external scenarios and both cable lengths**

| Case Studies | Cell | Configuration | SNR | SER | BER |
|---|---|---|---|---|---|
| Simulation | CA100AHA-ECM | 70 mm cable | 57.6 dB | 11.01 dB | ≈0 |
| | | 300 mm cable | 35.4 dB | 11.2 dB | ≈0 |
| | | 70 mm cable, noise | 27.1 dB | 10.8 dB | ≈0 |
| External Sensing | CA100AHA | 70 mm cable | 32.6 dB | 17.6 dB | ≈0 |
| | | 300 mm cable | 13.1 dB | 11.0 dB | $6.7 \times 10^{-11}$ |
| Internal Sensing | Customized Cell | 70 mm cable | 40.9 dB | 17.0 dB | ≈0 |
| | | 300 mm cable | 7.4 dB | 4.9 dB | $4.67 \times 10^{-4}$ |

The switching frequency is adjusted to $f_{\text{sw}} = 227$ kHz in the simulation, $f_{\text{sw}} = 262$ kHz during external sensing and $f_{\text{sw}} = 250$ kHz during internal sensing. In one configuration below, noise has been added to the demodulation signal in the simulation that emulates the system noise in the real setup.

be examined. The analysis was done with both batteries and the two connection cables, while the average battery current is zero (no load). To evaluate the SNR of the signal, the average power of the data pulses is calculated and divided by the superimposed noise generated by the converter ripple and the switching of the phase-sensitive detector. The probe noise captured during the measurements is compensated in the signal power calculations. The signal-to-error ratio is also employed to visualize the extent to which the pulses deviate from optimum pulses of the same amplitude. Consequently, the signal power is divided by the deviation signal power. Finally, the BER is calculated using:

$$\text{BER} = 0.5 \cdot \text{erfc}(\sqrt{\text{SNR}}) \qquad (5)$$

This analysis does not involve bit errors induced by timing mismatches, which result in crosstalk. Table 1 presents the SNR, SER, and BER achieved in simulations and experiments. The real BER thereby confirms

the low values in the table since no bit errors have been observed over the runtime of over 4 min.

## Discussion

During usage of the 70 mm long cable, the detection of the data is successful since no bit errors can be seen during a runtime of several minutes. The transmission is also tested under constant load, where the signal power does not decrease. Using the 300 mm cable instead results in high signal power degradation. It is expected that a proper adaptation of the switchable filter parameters to the cable impedance allows a better SNR under the usage of longer cables because the resonance between the output filter capacitor and the cable and battery's inductive behavior can be utilized to impact the current change by disturbing this resonance with the switchable filter. Accordingly, Fig. 5 can help to get an overview of possible sets of parameters.

As illustrated in Fig. 4a, the frequency with the highest admittance shift of the talkative battery matches almost the mathematical resonant frequency

**Fig. 9 | The potential of power modulation-based sensing in different applications to enable communication with the power converter.** *P* represents the power flow between a component and the converter, while $\hat{d}$ is the data transmitted from the component to the converter and back.

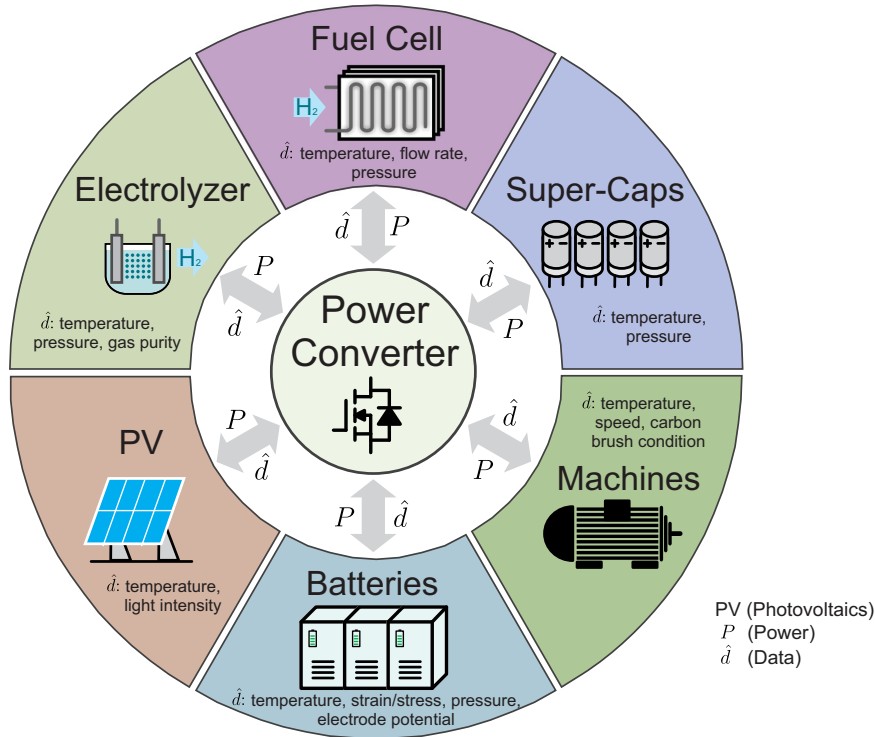

of the filter, which is 230 kHz. However, the switching frequency diverges for both batteries used in the case study, because small variations of the battery impedance, cable inductance $L_{tl}$, converter filter capacitance $C_F$, and inductance $L_F$ can shift the frequency point, at which the highest effect on the ripple is present. The strong effect of the resonance between $L_{tl}$, $\underline{Z}_c$ and $C_F$ is demonstrated in Fig. 5, in which a reduction of this resonant frequency by increasing $C_F$ results in a shift of the optimal point towards lower frequencies. This indicates that the required switching frequency to receive the data bits from the cell must be determined precisely after assembling the battery pack.

Focusing on the case where the 70 mm cable is attached to the battery and converter, in the experiments with the transmitter attached to the CA100AHA cell, the demodulation signal shifts with a voltage difference of 220 mV. Compared to the experiment with the customized cell, which showed a voltage difference of 555 mV, the signal power decreases a lot due to the low change of $\underline{Z}_{sc}$, which originates from the low impedance of the CA100AHA cell. When the long cable is used, the inductance $L_{tl}$ and resistance $R_{tl}$ between the cell and the converter increase. In the experiment with the CA100AHA cell, the demodulation signal shift reaches 30 mV instead of 24 mV when the customized cell is attached. This inverted performances of the battery cells is also reflected in the SNR metrics. The better performance in case of the CA100AHA battery cell might be generated by the enhanced resonance between $\underline{Z}_{dut}$ with $C_F$ since its impedance has an almost inductive nature. Once the corresponding resonance frequency is applied to the converter, it can be distorted using the switchable filter. In Fig. 5, this effect is located in region A. As demonstrated in the experimental findings, the activation of the switchable filter has been observed to result in a decrease in current ripple amplitude. This phenomenon can also be attributed to the distortion of the resonance process.

It can be stated that the successful transmission of temperature data from inside the cell using the LSK modulation strategy has been realized with a small amount of additional hardware, namely the switchable filter and the demodulation circuit. When a sufficiently low carrier frequency can be implemented, or a micro-controller with a high-speed analog-to-digital converter is available, it is possible to omit the analog demodulator by sampling the modulation signal directly. This could lead to more

straightforward implementation and reduced costs for temperature sensing systems.

## Conclusion

A low-cost sensing methodology utilizing power electronics converters to collect critical battery data was introduced, conceptualized, and experimentally validated. Cost-effective internal and external temperature sensing at the cell level can be achieved for large-format cells, eliminating the need for a dedicated communication device. The communication signal frequency is decoupled from the DC power signals used for charging and discharging the talkative battery. Therefore, once suitable filters have been tuned to communicate with the converter, they can be applied to batteries of different capacities without the need for recalibration. This simple, straightforward method solely depends on the availability of electrical connections, and the implementation of internal and external sensing is basically the same. While this paper focuses on temperature sensing to ensure the safety of lithium-ion batteries, the methodology can be generalized to many other sensor technologies and diverse environments. The proposed method opens up many possibilities for communication in challenging environments, as encountered in most energy applications. Figure 9 shows some energy system components that are interfaced with power electronic converters. These components can benefit from the proposed method to realize a communication channel with the power converter. It can also function as a secondary communication channel in situations where the primary communication link might fail.

## Methods
### EIS

EIS was used to analyze the cell impedance. For its application, an AC current with variable frequency is applied to the battery, and the cell voltage is measured. The RMS value of the applied current is $I_c = 0.5$ A. According to the formula

$$\underline{Z}_c(\omega) = \frac{\underline{u}_c(\omega)}{\underline{i}_c(\omega)} \tag{6}$$

with $\underline{u}_c$ and $\underline{i}_c$ as the complex phasors of voltage and current at the battery cell, the impedance $\underline{Z}_c$ is calculated as a function of the angular frequency $\omega$[45]. EIS has been applied for 23 different SOC's by alternating EIS procedure and short current pulses for SOC incrementation several times.

## Demodulation

For demodulation, a PSD has been implemented[46]. As shown in Fig. 10, the current $i_{tl}$ is measured and the low-frequency components are removed. This is realized with a single RC high-pass filter with a time constant of $\tau_3 = 81$ ms. The high-pass filtered signal $u_{sens.hpf}(t)$ is fed into an inverting and a non-inverting amplifier. The combined gain of the current sensor and amplifiers is $0.59$ V A$^{-1}$. A multiplexer swaps between those two signals according to a control signal, which is provided as a PWM from the TMS320F28379D micro-controller and has the same frequency as the converter PWM but can hold a different phase.

To express this process mathematically, a signal is defined as the demodulation PWM, which is according to

$$\xi(t) = 2s_d(t) - 1 = \begin{cases} 1 & \text{for } t \bmod T < T/2 \\ -1 & \text{for } t \bmod T \geq T/2 \end{cases} \tag{7}$$

switching between $-1$ and $1$. In the subsequent analysis, the phase delay, denoted with $\phi_{dem}$, is set to zero to facilitate the analysis. The output of the multiplexer can be expressed as

$$u_{mx}(t) = G u_{sens.hpf}(t)\xi(t) \tag{8}$$

It can be analyzed using the Fourier series[47]. The Fourier coefficients of $\xi(t)$ are accordingly:

$$\Xi(k) = \frac{1}{T}\int_{-T/2}^{T/2} \xi(t) \exp\left(-\frac{2\pi j t}{T}k\right) dt =$$
$$\frac{\left[\exp\left(-\frac{2\pi j t}{T}k\right)\right]_0^{T/2} - \left[\exp\left(-\frac{2\pi j t}{T}k\right)\right]_{-T/2}^0}{-2\pi jk} = \frac{2(k \bmod 2)}{\pi jk} \tag{9}$$

The separation into its sinusoidal components in the complex domain results in

$$\xi(t) = \sum_{k\in\mathbb{Z}} \Xi(k) \exp\left(\frac{2\pi jk}{T}t\right) = \sum_{k\in\mathbb{Z}} \frac{2}{\pi j(2k-1)} \exp\left(\frac{(2k-1)\pi j}{T}t\right) \tag{10}$$

and can be transferred back to the real numbers:

$$= \sum_{k\in\mathbb{Z}} \frac{2}{\pi(2k-1)} \sin\left(\frac{(2k-1)\pi}{T}t\right) = \frac{4}{\pi}\sum_{k=1}^{\infty} \frac{\sin\left(\frac{(2k-1)\pi k}{T}t\right)}{(2k-1)} \tag{11}$$

Finally, we get the multiplexer output:

$$u_{mx}(t) = G u_{sens}(t)\frac{4}{\pi}\sum_{k=1}^{\infty} \frac{\sin\left(\frac{(2k-1)\pi k}{T}t\right)}{(2k-1)} \tag{12}$$

In signal theory, the product of two time-series functions results in a convolution in the frequency domain. Since the PWM signal consists in the frequency domain of a comb of decreasing unit impulses, we will get a wide distribution of shifted and weighted spectra of $u_{sens}$ around all peaks of the PWM spectrum after the convolution. If we now assume that $u_{sens}$ contains only a ripple that has the amplitude $D$, equal time constant $T$ and relative phase shift to the PWM of $\theta$, we can reformulate the equation.

$$u_{mx}(t) = GM\cos\left(\frac{2\pi t}{T}+\theta\right)\cdot\frac{4}{\pi}\sum_{k=1}^{\infty}\frac{\sin\left(\frac{(2k-1)\pi t}{T}\right)}{(2k-1)} =$$
$$G\frac{4}{\pi}\sum_{k=1}^{\infty}\frac{\frac{M}{2}\left(\sin\left(\frac{(2k-1)2\pi}{T}t+\frac{2\pi t}{T}+\theta\right)+\sin\left(\frac{(2k-1)2\pi}{T}t-\frac{2\pi t}{T}-\theta\right)\right)}{(2k-1)} =$$
$$G\frac{4}{\pi}\sum_{k=1}^{\infty}\frac{\frac{M}{2}\left(\sin\left(\frac{4\pi k}{T}t+\theta\right)+\sin\left(\frac{(2k-2)\pi}{T}t-\theta\right)\right)}{(2k-1)} \tag{13}$$

It can be seen that for $k = 1$ a DC quantity exists. After filtering out all additional harmonics, which have frequencies equal or higher than the switching frequency, the following expression remains:

$$u_{dem} = G\frac{4}{\pi}\frac{M}{2}\sin(-\theta) \tag{14}$$

The signal $u_{dem}$ contains a dependence on the magnitude M and on the phase difference $\theta$. The low-pass filtering is realized with two RC low-pass filters with time constants of $\tau_1 = \tau_2 = 47$ μs. The sampling period is 100 μs. After analog to digital conversion, the signal is then again filtered digitally and compared to a reference in the micro-controller to detect the starting bit. For the starting bit detection, the filter is realized as a simple IIR filter. Once a start bit is detected, 20 acquisitions of the signal are performed during the period, where a single bit is located. Afterward, the average value of those acquisitions is compared with the threshold to detect the bits. The threshold gets dynamically adapted between the level for a zero and the level for a one. Then again, 20 samples are detected for the next bit.

**Remark:** Although the carrier generation is not done by the miniaturized transmitter, its power consumption cannot be assumed to be 0 W because the ATmega328p is still powered, and the voltage divider for reading the NTC also draws current. To have no power consumption when the battery is not in use, zero power wake-up techniques can avoid a slow discharge of the cell[48]. However, the total discharge rate of large format cells

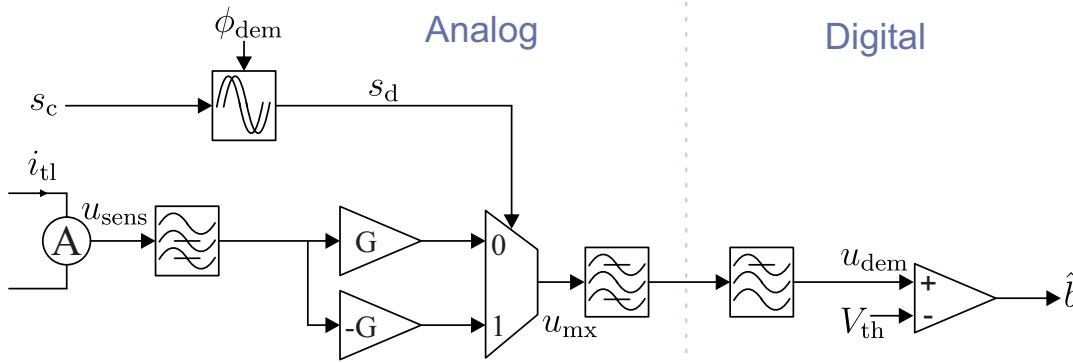

$i_{tl}$(Tallkative Battery Current)  $u_{sens}$(Current Sensor Output)  $u_{mx}$ (Multiplexer Output)  $s_c$ (Pulse Width Modulation)
$\phi_{dem}$(Phase Shift)  $u_{dem}$(Demodulated Signal)  $V_{th}$(Bit Threshold)  $s_d$(Shifted Pulse Width Modulation)  $\hat{b}$ (Data)

**Fig. 10 | Schematic of the implanted Phase Sensitive Detector.** The analog section contains the current sensing, PWM controlled multiplexing and filtering. After analog to digital conversion, the signal gets filtered again and compared with threshold value to retrieve the bit stream.

induced by the power consumption of the ATmega328P is negligibly low, making zero power wake-up techniques more reasonable for smaller cell formats.

## Data availability

The datasets generated by the experiments, as well as the transmitter and receiver microcontroller programming code, can be provided on request. All other data that support the findings of this study are also available from the corresponding author upon reasonable request

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

## Author contributions

Johannes Diers implemented the LSK-based transmitter and receiver, transforming the regular battery into a "talkative" battery. He also performed the experiments presented in the paper and created the mathematical analysis of the LSK principle. Hamzeh Beiranvand developed the core concept of the talkative battery, incorporating the filter characteristics and the idea of using the power converter ripple as a carrier. He also supervised the writing process and provided guidance on creating the figures.

## Funding

## Competing interests

The authors declare no competing interests.
