## [Transparent Peer Review file · Communications Engineering]

Talkative Battery: Super-Safe Batteries with Power-Modulation Based Internal and External Sensor Data Collection

Corresponding Author: Dr Hamzeh Beiranvand

Version 0:

Reviewer comments:

Reviewer #1

(Remarks to the Author)

This manuscript presents a novel concept for low-cost internal and external sensing in lithium-ion batteries by exploiting power-modulation-based communication using the Load-Shift Keying (LSK) principle. The proposed method enables data transmission from embedded sensors within large-format Li-ion cells to the power converter without additional wiring or dedicated communication transceivers. The work demonstrates solid theoretical grounding, including impedance modeling via Electrochemical Impedance Spectroscopy (EIS), circuit-level analysis, and both simulation and experimental validation. Results obtained with 100Ah LFP cells and custom-made cells convincingly prove the feasibility of the concept. However, while the paper is technically sound and the contribution is promising, the novelty is not sufficiently distinguished from existing power-line or converter-modulated communication methods, and the experimental validation lacks comprehensive quantitative data on communication robustness, stability, and performance limits. In addition, several figures and explanations could be clarified to improve readability. In my opinion the paper is suitable for publication after a major revision addressing the following points.

1. The concept of using converter-induced ripple for data transmission has been discussed in prior “talkative power conversion” and PLC-related studies. Please clarify the specific innovation of this work (e.g., bidirectional path, embedded sensor integration, reduced hardware, or signal modulation method).
2. Provide quantitative communication performance metrics. Only SNR, SER, and BER are presented for limited conditions. Please include such as Data rate and effective bandwidth; Transmission stability over time and under varying load currents. These parameters are essential to assess real-world applicability.
3. The PWM frequency used in experiments (≈ 262 kHz) deviates from the simulated resonance (200 kHz). Please analyze this discrepancy quantitatively and discuss parameter sensitivities.
4. The Discussion claims that the method can be extended to other sensor types (strain, pressure, gas detection). Please provide corresponding further analysis and explanation or concept demonstration.
5. Figures 3 and 7 are very dense and hard to follow. Please redraw them with clearer signal-flow paths, consistent labels, and increased font sizes.
6. Could you briefly elaborate on the current set limitations, the prospects of system-level integration and the future work direction?
7. Replace repeated “converter PWM frequency” with “switching frequency” for clarity.
8. Specify units and conditions in Table 1.
9. Improve English fluency in some sentences.

Reviewer #2

(Remarks to the Author)

The paper proposes and validates a novel, low cost method for collecting internal and external sensor data from lithium-ion battery (LIB) cells using power modulation-based communication, specifically employing Load Shift Keying (LSK). This is definitely an interesting research endeavor in the LIB sensing space. This reviewer has a few concerns and comments regarding this work that the authors could perhaps address:

1. The authors assert that the proposed hardware to enable individual cell sensing is low-cost. However, the said low-cost is

not quantified in the paper or compared to current state-of-the-art (SOA). Please note that the SOA is individual cell voltage sensor, string-level current sensor, and at least a few thermocouples to capture max./min. surface temperatures for each module.

2. Besides introduction of power converter for each cell can still significantly increase cost and complexity for the overall system based on system size. For example, there could still be 1000s of large format cells in multi-MW utility-scale battery energy storage systems.

3. While the advantage of using this technology for internal temperature sensing compared to estimation-based methods is clearly justified, it is not clear how easily the hardware necessary for internal sensing along with all the power electronics proposed by the authors can be manufactured through hitherto well-established manufacturing practices for LIBs. Even in the study here, internal sensing is not truly internal. Additional comments or study is needed to justify that.

4. What is the power consumption of the total external hardware as a fraction of the cell Ampacity? Does that consumption scale linearly with the number of cells in a battery pack? I am trying to get a sense of the auxiliary load on a system owing to this method of sensing.

5. Is there a limit to the total cell current with this sensing method? The average current used in the experiments are of relatively lower C-rates compared to what LFP and some other LIB chemistries can handle.

6. Figure 8 is very crowded and the texts are too small to read. Additionally, adding a de-encoded temperature measurement figure would be helpful.

Hopefully, these comments help the authors in further refining the paper.

Reviewer #3

(Remarks to the Author)

The authors presented a novel method to communicate measurements of the internal temperature of Li-Ion cells, but without additional wiring/cabling as is the case with some other methods proposed in literature. The method makes use of an inductor/capacitor combination that is switched in and out across the Li-Ion cell to change the equivalent impedance seen by the power converter. This, if well designed, leads to the ripple, originating from the switching of the power converter, to change that again provides a method to modulate the serial communication from the cell.

The authors went through great effort to analyze the method and to develop the underlying theory. They then go on to demonstrate the method both through simulation and experimentally. They also investigated and demonstrated different cases of cable length and the influence on the signal to noise ratio (SNR), and the associated SER and BER.

There are a few challenges to this method as applied to Li-Ion batteries (which is the primary application of the method):

In the case of Li-Ion batteries, where multiple cells are switched in series to obtain a higher voltage battery, it is not clear what the success of this method will be. Judging from the deterioration of the SNR for a longer cable length reported in the paper, the additional series impedance of multiple other cells could make the method unusable. That is, modulation on the ripple will be undetectable once a cell is switched in series with multiple other cells.

A further uncertainty is how widely applicable the method is due to the wide range of power electronic converters that are available. It is quite possible that a Li-Ion cell or a battery consisting of several cells in series could work with one converter and not with another. Since the intention is presumably to incorporate the measurement and modulation circuitry in the cell, prior knowledge of the converter is not possible. So how will one ensure that a cell will be able to communicate irrespective of the arrangement in which it is installed? Here it seems that some of the other methods proposed in literature has a clear advantage.

In summary, the authors presented a novel power-modulation technique including a comprehensive development of the underlying theory and they demonstrated the method using both simulations and experimentally. And while there are practical challenges as highlighted above, I am of the opinion that this work contributes to the body of knowledge. I therefore propose that the paper be accepted. I do however urge the authors to address at least in the discussion at the end of paper the two challenges I raised above.

Version 1:

Reviewer comments:

Reviewer #1

(Remarks to the Author)

I have reviewed the revised manuscript and am satisfied with the changes. The authors have addressed all my previous concerns. I recommend that the paper be accepted for publication.

Reviewer #2

(Remarks to the Author)

Dear Authors,

Thank you for providing detailed clarifications on my earlier comments. While I completely agree with your responses to my other comments, I am not convinced with the low cost claim (comment #1) of your proposed technology. The cost analysis presented is a bit convoluted. It would have been easier to follow if you had provided a basic bill of material (BOM) for a state-of-the-art battery pack with corresponding costs and a BOM for the same battery pack now containing your sensing technology. If your claim is correct, the latter BOM should show reduced total cost compared to the former.

Moreover, while it is true that with larger format cells, the quantity of cells per module might decrease, I am not sure if that directly translates to less temperature sensors. It is well known that temperature heterogeneity is more prominent with larger format (prismatic) cells than smaller cylindrical cells. To ensure better safety and avoid liability, a conservative design would consider multiple temperature sensing per cell for larger format cells.

Other than the comment above, I fully agree with your other responses and corresponding changes made in the revised manuscript.

Version 2:

Reviewer comments:

Reviewer #2

(Remarks to the Author)

Thank you for performing the due diligence in support of your low cost solution claim.

1 Reviewer Responses

1.1 Reviewer: 1

This manuscript presents a novel concept for low-cost internal and external sensing in lithium-ion batteries by exploiting power-modulation-based communication using the Load-Shift Keying (LSK) principle. The proposed method enables data transmission from embedded sensors within large-format Li-ion cells to the power converter without additional wiring or dedicated communication transceivers. The work demonstrates solid theoretical grounding, including impedance modeling via Electrochemical Impedance Spectroscopy (EIS), circuit-level analysis, and both simulation and experimental validation. Results obtained with 100Ah LFP cells and custom-made cells convincingly prove the feasibility of the concept. However, while the paper is technically sound and the contribution is promising, the novelty is not sufficiently distinguished from existing power-line or converter-modulated communication methods, and the experimental validation lacks comprehensive quantitative data on communication robustness, stability, and performance limits. In addition, several figures and explanations could be clarified to improve readability. In my opinion the paper is suitable for publication after a major revision addressing the following points.

We appreciate your time and effort reviewing our manuscript. We have addressed your comments one by one in the following, which increased the overall quality of the paper.

1. The concept of using converter-induced ripple for data transmission has been discussed in prior “talkative power conversion” and PLC-related studies. Please clarify the specific innovation of this work (e.g., bidirectional path, embedded sensor integration, reduced hardware, or signal modulation method).

Response:

The innovation of this work lies in the proposed signal modulation method LSK and the additional resonance filter used in battery side to realize the LSK. The method presented in this paper can be indeed classified as a talkative power conversion (TPC) method. In the current TPC technologies, the **data is modulated within the power converter**, for example using the pulse width modulation (PWM) phase or frequency. However, we propose load shift keying, and **the data is modulated outside of the converter**. The innovation lies in implementing the load shift keying approach to measure battery temperature. The benefits of the proposed method include low material requirements (e.g. wiring), low costs, and very low transmitter power consumption.

- The title is improved to better reflect the talkative power conversions: new title: **Talkative Battery: Super-Safe Batteries with Power-Modulation based Internal and External Sensor Data Collection**
- The abstract and the overall text have been adapted to the improved title.

- Following change has been added to the paper to better clarify the differences: on page 2 line 73-77 and on page 3 line 84-86.

2. Provide quantitative communication performance metrics. Only SNR, SER, and BER are presented for limited conditions. Please include such as Data rate and effective bandwidth; Transmission stability over time and under varying load currents. These parameters are essential to assess real-world applicability.

Response:

In the experiments presented in the paper, a data rate of 232 bit s^{-1} and an effective bandwidth of 153 bit s^{-1} , which incorporates the transmission and reception time, is used. In order to illustrate the transmission stability over time with different load currents, the following experiment is conducted. The false bit flips were counted, while the battery has been charged with a current of 18A for 2617s, as depicted in figure ???. Afterwards, the battery rests for 2600s. The battery cell used is a 48Ah NMC pouch cell, which exhibits an even lower impedance compared to the CA100AHA LFP cell. The temperature is tracked over the time span of 5455s and bit flips in the first 9 positions of every transmitted message are tracked. Channel coding has been employed to bypass messages containing errors in the initial 9 bits. These errors are identified through a comparison of the current message with previous messages, thereby observing high instantaneous variations in the transmitted ADC value. During the operation, in total 218 bit errors have been detected. Since every 43ms a new message gets transmitted, it can be stated, that in total $\frac{5455s}{43ms} 10\text{bit} = 1.27\text{Mbit}$ have been sent. Hence, the effective BER is $\frac{218\text{bit}}{1.27\text{Mbit}} = 0.17\%$. As stated in the paper, the battery impedance has a strong impact on the magnitude of the impedance shift, which affects also the BER. However, among all three battery types, the 48Ah battery cell used for the experiment shown in figure ??? exhibits the lowest impedance and thereby the highest BER. For the two cells discussed in the paper, no bit errors appeared during a runtime of 4 min, which has been added in line 305 to 307.

Additional metrics have been added to the paper on page 10 line 288-292.

3. The PWM frequency used in experiments ($\approx 262 \text{ kHz}$) deviates from the simulated resonance (200 kHz). Please analyze this discrepancy quantitatively and discuss parameter sensitivities.

Response:

We adjusted the LC filter parameters in the the simulation based on our measurements and the new resonance frequency of simulations is 230kHz. The deviation of the switching frequency is induced by several circumstances:

- Deviation between the theoretical value of L_f and C_f versus the built in components. Subsequent measurements of the parameters show $L_f = 0.8 \mu\text{H}$ and $C_f = 600 \text{ nF}$, which results in a resonant frequency of 230kHz.

- Contributions from the cable inductance L_{tl} . As depicted in the figure 5 of the original paper, due to the resonance between the cable inductance L_{tl} and the converter filter capacitor C_F , the frequency with highest current ripple difference ΔH_{CLSK} can shift significantly away from the resonant frequency (area A and B in the figure). Since very small changes in the cable assembling can affect the value of L_{tl} , it becomes clear that during the setup assembling, the inductance of the transmission cable was lower compared to the value measured with the impedance analyzer.
- The inductance of L_F is constructed using a ferrite coil, which renders it nonlinear. This, in turn, has an effect on the resonant behavior of the system.

On page 11 line 334-342 the aspect of the frequency sensitivity on L_{tl} , C_F , and L_F has been added, and a reference back to the figure 5, which is addressing the dependency of the optimal frequency on the resonance between the connection cable and the converter filter capacitor, is implemented

4. The Discussion claims that the method can be extended to other sensor types (strain, pressure, gas detection). Please provide corresponding further analysis and explanation, or concept demonstration.

Figure 1: Concept demonstration of connecting several sensor types into the proposed talkative battery (TB).

Response:

Figure 1 shows the concept demonstration of connecting different types of sensors to the same resonant filter for LSK modulation. The sensing part and transmission part of the sensor system are fully decoupled. Any of these sensor types provides an analog output that can be converted into a bit stream by analog-to-digital conversion. Accordingly, the implementation of another sensor type, like strain, pressure, or gas, will not differ from the integration of the temperature sensor.

5. Figures 3 and 7 are very dense and hard to follow. Please redraw them with clearer signal-flow paths, consistent labels, and increased font sizes.

Response:

The new figures will be improved according to the reviewers' suggestions.

- Figure 3 has been improved by increasing the font sizes of the axis description, the explanation, and designators.
- Figure 7 has been reduced by removing the microcontroller and the font size of the designators has been increased. Furthermore, the figure was split into two sub-figures (a) and (b), showing internal and external experimental setups separately.

We believe that the new figures are much simpler and illustrative, and the reader can find out the main idea immediately at a glance.

6. Could you briefly elaborate on the current set limitations, the prospects of system-level integration and the future work direction?

Response:

We would address the limitations and prospective developments in the following points:

- **Limitations:** Since this is the very first of our experiments and experiences on the talkative batteries, many aspects are still unknown to us, including the limitations. However, one can imagine that some limitations might arise from the limited switching frequency of the converter due to increased switching losses. Therefore, there will be an upper limit on the data rate, and further research is needed in this direction. A second limitation might be related to the integration of electronics inside the battery cell if internal sensing is targeted. In such a situation, the electronic circuit elements might be influenced by the electrolyte, and therefore, suitable insulation between the LC filter and the electronics might become a limitation.
- **Future prospective:** The present study focuses on the data transmission from a single cell to a power converter. However, future research will explore the implementation of this communication approach in serial multiple cell battery packs. As stated in the paper, the dependency of the impedance shift on the resonant frequency of the switchable filter is a significant aspect of the system that requires further exploration for multi-cell communication. Nevertheless, the limitation in cell numbers has also not yet been subjected to evaluation. Figure ?? illustrates the conceptual idea of multicell data reception with the proposed strategy.

7. Replace repeated “converter PWM frequency” with “switching frequency” for clarity.

Response: It has been implemented as suggested.

8. Specify units and conditions in Table 1.

Response: The units and conditions have been added to the description of the table.

9. Improve English fluency in some sentences.

Response:

We read the paper several times carefully and checked the writing with DeepL. We are confident that the new version is substantially improved in terms of writing.

1.2 Reviewer: 2

The paper proposes and validates a novel, low cost method for collecting internal and external sensor data from lithium-ion battery (LIB) cells using power modulation-based communication, specifically employing Load Shift Keying (LSK). This is definitely an interesting research endeavor in the LIB sensing space. This reviewer has a few concerns and comments regarding this work that the authors could perhaps address:

Thank you very much for the insightful comments. We have addressed all the comments in the following.

1. The authors assert that the proposed hardware to enable individual cell sensing is low-cost. However, the said low-cost is not quantified in the paper or compared to current state-of-the-art (SOA). Please note that the SOA is individual cell voltage sensor, string-level current sensor, and at least a few thermocouples to capture max./min. surface temperatures for each module.

Response:

Although we suggest that the perfect solution is to have sensors in each cell, the proposed method can be used to insert sparse sensors exactly as in present SOA but without the need for dedicated communication systems. Figure 2 demonstrates the concept of only equipping a limited number of cells in the battery pack with temperature sensors and switchable filters. The present SOA takes into account individual voltage sensors for the battery cells, a string-level current sensor, and several thermocouples per module. This assertion also pertains to battery packs that contain large cells. Whilst the quantity of cells per module is reduced when larger cells are employed, the number of voltage sensors is reduced, thereby lowering the cost. Concurrently, the number of temperature sensors required for the proposed individual cell supervision is correspondingly diminished and wiring costs are totally avoided. However, the number of temperature sensors per module remains constant within the SOA. It is therefore evident that the proposed approach is cost-efficient for battery packs that possess larger batteries.

The figure has been taken out of the public version of the reviewer correspondence.

To demonstrate how the costs are changed vs the cell size we performed numerical analysis assuming a fixed battery volume of 1 m^3 , voltage of 400 V , cell voltage

of 3.5 V and cell energy density of 500 WhL^{-1} . A per unit cost system is used to make the comparison meaningful: voltage sensor are $c_1 = 0.3$, current sensor $c_2 = 0.5$, temperature sensor $c_3 = 0.2$, and wiring of a sensor $c_4 = 0.2$. The cost of the communication device is $c_5 = 0.2$. Figure 2 in this document illustrates the lower sensor costs for large cell sizes while using the talkative battery (TB) approach, which involves applying a temperature sensor to every fifth battery cell. The figure 2a) demonstrates the case, that cells are connected in series to strings and the strings are parallelized, each having a current sensor while voltage sensors are attached to each cell. In the opposite case, considered in 2b), cells are connected parallel to packs with a single voltage sensor per pack. These packs are connected serially to reach the desired voltage. Such a parallel pack configuration of cells requires only a single talkative battery transmitter, if a temperature sensor is implemented into one or more of the parallel cells.

Figure 2: **Battery pack design for a fixed volume of 1 m^3 , voltage of 400 V and cell energy density of 500 WhL^{-1} .** a) Cells are connected serially to strings, which are parallelized, or b) cells are parallel connected into packs with a single transmission unit and voltage sensor per pack. Those packs are serially connected to reach the desired voltage, which also requires just a single current sensor. The state-of-the-art (SOA) example includes fixed 50 sensors, but the talkative battery (TB) example includes every fifth cell with a temperature sensor. Costs per unit for the voltage sensor are $c_1 = 0.3$, current sensor $c_2 = 0.5$, temperature sensor $c_3 = 0.2$, wiring of a sensor $c_4 = 0.2$, and transmission unit $c_5 = 0.2$.

2. Besides introduction of power converter for each cell can still significantly in-

crease cost and complexity for the overall system based on system size. For example, there could still be 1000s of large format cells in multi-MW utility-scale battery energy storage systems.

Response: The author does not recommend interconnecting all cells in large-scale battery packs, each with a converter attached; rather, the current ripple of a single converter with varying frequencies can enable data transmission from all cells within a battery pack of serially connected cells. A follow-up paper will discuss the application of this methodology to multiple interconnected cells. The corresponding technical realization is illustrated in 5. **The figure has been taken out of the public version of the reviewer correspondence.**

3. While the advantage of using this technology for internal temperature sensing compared to estimation-based methods is clearly justified, it is not clear how easily the hardware necessary for internal sensing along with all the power electronics proposed by the authors can be manufactured through hitherto well-established manufacturing practices for LIBs. Even in the study here, internal sensing is not truly internal. Additional comments or study is needed to justify that.

Response:

The paper's objective is to demonstrate the functionality of the novel transmission technology and not the temperature sensor technology itself. As you mentioned, a true sensing element integration requires modifying at least the battery winding construction in the production line, which is out of the focus of the paper. Nevertheless, the variety of internal temperature measurement approaches can be reviewed in [1], in which it is stated that RTDs and thermocouples are the most common sensor types used for internal temperature detection of commercial batteries. Furthermore, new integrated sensor types for batteries are developed, which can easily be integrated into the battery manufacturing process, since they are directly applied on the battery components [2]. This indicates that the integration of sensors, especially for temperature, will not pose a big challenge in Li-ion battery manufacturing, and smarter sensor integration approaches are in development. However, the transmitters manufactured for the experiments still require modification to further reduce their size to be more easily implementable in prismatic battery cell cases. Since such a large-scale battery pack is usually interconnected with power electronics and a current sensor, the only requirement for the power electronics is the requested ripple, which is typical for power converters and does not restrict or change their manufacturing process.

4. What is the power consumption of the total external hardware as a fraction of the cell Ampacity? Does that consumption scale linearly with the number of cells in a battery pack? I am trying to get a sense of the auxiliary load on a system owing to this method of sensing.

Response:

For the charge of N cells connected in series a single external power converter will be required, and that this will not be scalable in accordance with the number

of sensors. In order to provide a more precise indication of the hardware's power consumption, the measurement is divided into the following groups:

- **Converter control:** The consumption of the signal processing hardware is dominated by the microcontroller (TMS320f28379D) and the current sensor, which is in total 1.15 W (230 mA at 5 V).
 - **Converter electronics:** The gate driver and over-current and over-voltage protection consume 2.5 W (500 mA at 5 V).
 - **Converter main switch and LC filter losses:** For this measurement, the converter has been operated while connected to the battery, but without any connection to the main power supply. During this operation, the converter consumes 278 mA from the battery, which originates from switching losses, and the bleeding resistors of the capacitors on the primary side of the half bridge. The discharge C-rate is 0.0028. However, it should be highlighted that these components, such as the converter and the current sensor, are key components to ensure a controlled energy flow between the battery cell and potential energy sources or sinks, which is why they can not be neglected. Typical efficiency of the converter with up to date semiconductor technologies reaches above 98 %.
 - **Internal sensor:** At 3 V battery voltage, its consumption is 12 mW. In case of the cell being unused, zero power wake-up techniques should be applied to reduce even more the internal sensors consumption.
5. Is there a limit to the total cell current with this sensing method? The average current used in the experiments is of relatively lower C-rates compared to what LFP and some other LIB chemistries can handle.

Response:

The power and communication are fully decoupled in the frequency domain, and there should be minimal effects from the load level. However, the limitation is mainly related to the adopted devices in the realization of the setup, such as the current sensor, which, in this instance, is the ACS732KMATR-65AB-T manufactured by Allegro. It has a current limitation of 65 A.

6. Figure 8 is very crowded and the texts are too small to read. Additionally, adding a de-encoded temperature measurement figure would be helpful.

Response:

Figures are improved with a larger font size and improved structure. Figure ?? in this document depicts the transmitted temperature sensor data in de-encoded form, while the attached battery has been charged first and subsequently rested. The battery used is a 48 Ah NMC battery cell with pouch cell format, which showed a strong thermal reaction sensed by the attached sensor. **The figure has been taken out of the public version of the reviewer correspondence.**

Hopefully, these comments help the authors in further refining the paper.

We appreciate your feedback. The comments helped us to improve the revised version significantly.

1.3 Reviewer: 3

The authors presented a novel method to communicate measurements of the internal temperature of Li-Ion cells, but without additional wiring/cabling as is the case with some other methods proposed in literature. The method makes use of an inductor/capacitor combination that is switched in and out across the Li-Ion cell to change the equivalent impedance seen by the power converter. This, if well designed, leads to the ripple, originating from the switching of the power converter, to change that again provides a method to modulate the serial communication from the cell.

The authors went through great effort to analyze the method and to develop the underlying theory. They then go on to demonstrate the method both through simulation and experimentally. They also investigated and demonstrated different cases of cable length and the influence on the signal to noise ratio (SNR), and the associated SER and BER.

Response: We would like to thank you for the effort you put into reviewing our manuscript and for your suggestions to improve the paper.

There are a few challenges to this method as applied to Li-Ion batteries (which is the primary application of the method):

In the case of Li-Ion batteries, where multiple cells are switched in series to obtain a higher voltage battery, it is not clear what the success of this method will be. Judging from the deterioration of the SNR for a longer cable length reported in the paper, the additional series impedance of multiple other cells could make the method unusable. That is, modulation on the ripple will be undetectable once a cell is switched in series with multiple other cells.

Response:

We have already dealt with this challenge of scaling the number of cells connected in series as shown in figure ???. The setup of the four serially connected talkative batteries is demonstrated in figure ??b. We observed that the increased impedance due to connecting cells in series leads to lower signal power at converter terminals and therefore it becomes challenging to decode the data by only one current sensor. To alleviate the challenge of scaling, we suggest to use the voltage measurement at cell level. Voltage sensing is a necessary part for battery management system for cell balancing. The voltage measurement at cell level is not influenced by the number of connected cells and is scalable to any number of cells. The application of the proposed method in the case of multiple serially connected cells will be discussed in a subsequent paper.

The figure has been taken out of the public version of the reviewer correspondence.

A further uncertainty is how widely applicable the method is due to the wide range of power electronic converters that are available. It is quite possible that a Li-Ion cell or a battery consisting of several cells in series could work with one converter and not with another. Since the intention is presumably to incorporate the measurement and modulation circuitry in the cell, prior knowledge of the converter is not possible. So how will one ensure that a cell will be able to communicate irrespective of the arrangement in which it is installed? Here it seems that some of the other methods proposed in literature has a clear advantage.

Response:

To generalize the method for other power converter topologies, we need to pay attention the way the ripple amplitude and frequency can be altered. The frequency of ripple can be controlled by switching frequency. For example in the converter used in this paper switching frequency is equal to the ripple frequency $f_{rip} = f_{sw}$. The method can be applied to a wide range of converters with similar principles to buck converter, immediately. However, for some other converter as you pointed out, might be not straight forward. For example, in isolated DC-DC converters such dual active bridges (DAB) or resonant converter types (LLC and CLLC converters), the switching frequency is affecting the power transfer and voltage control with $f_{rip} = 2f_{sw}$, respectively. Therefore, designing the switchable resonant filter that could provide a wider bandwidth can be approached. Such a topologies demand for higher bandwidth and frequency division multiple access will not be suitable for battery pack level, instead, time division multiple access can be adopted. We believe that further research should be conducted to unfold the full potential of the proposed approach. we hope that we can address some of the points in our future works.

In summary, the authors presented a novel power-modulation technique including a comprehensive development of the underlying theory and they demonstrated the method using both simulations and experimentally. And while there are practical challenges as highlighted above, I am of the opinion that this work contributes to the body of knowledge. I therefore propose that the paper be accepted. I do however urge the authors to address at least in the discussion at the end of paper the two challenges I raised above.

We are delighted by your feedback and thank you for your helpful comments on our manuscript.

References

1. Jinasena, A. *et al.* Online Internal Temperature Sensors in Lithium-Ion Batteries: State-of-the-Art and Future Trends. *Frontiers in Chemical Engineering* **4**, 804704. ISSN: 2673-2718. <https://www.frontiersin.org/articles/10.3389/fceng.2022.804704/full> (2025) (Feb. 16, 2022).

Dr. Hamzeh Beiranvand

Group Leader Battery Systems, Chair of Power Electronics

hab@tf.uni-kiel.de | +49 431 880 0111 | www.beiranvand.org

2. Paljk, T. *et al.* Integrated sensor printed on the separator enabling the detection of dissolved manganese ions in battery cell. *Energy Storage Materials* **55**, 55–63. ISSN: 24058297. <https://linkinghub.elsevier.com/retrieve/pii/S2405829722006316> (2024) (Jan. 2023).

1 Reviewer Responses

1.1 Reviewer: 1

I have reviewed the revised manuscript and am satisfied with the changes. The authors have addressed all my previous concerns. I recommend that the paper be accepted for publication.

We are grateful for your recommendation and thank you for your suggestions in the review process.

1.2 Reviewer: 2

Thank you for providing detailed clarifications on my earlier comments. While I completely agree with your responses to my other comments, I am not convinced with the low cost claim (comment technology). The cost analysis presented is a bit convoluted. It would have been easier to follow if you had provided a basic bill of material (BOM) for a state-of-the-art battery pack with corresponding costs and a BOM for the same battery pack now containing your sensing technology. If your claim is correct, the latter BOM should show reduced total cost compared to the former.

Moreover, while it is true that with larger format cells, the quantity of cells per module might decrease, I am not sure if that directly translates to less temperature sensors. It is well known that temperature heterogeneity is more prominent with larger format (prismatic) cells than smaller cylindrical cells. To ensure better safety and avoid liability, a conservative design would consider multiple temperature sensing per cell for larger format cells.

Other than the comment above, I fully agree with your other responses and corresponding changes made in the revised manuscript.

We are pleased that our clarifications and corrections have addressed the majority of the inquiries raised. We agree that a simple bill of materials (BOM) of a battery pack with the state of the art temperature supervision and with the proposed methodology will make our low-cost claim tangible. We furthermore agree that over a certain cell size, temperature heterogeneity of a single cell should be incorporated, which requires multiple temperature sensors per cell. With the 100Ah AC100AHA battery used in our paper, the location of the sensor can already impact the detected temperature trend significantly during battery operation. This is why we considered more than one temperature sensor for a single cell in the BOM discussed subsequently. Using several sensors per cell is even more cost-effective with the proposed solution, since more sensors can be managed by the same microcontroller inside(outside) the cell with transmission over the common LSK-transmitter.

The BOM is divided into the battery pack components that are not part of the sensing unit, which can be seen in table 1, and three further tables are dedicated to temperature measurement. The conventional battery temperature measurement using wires to connect all sensor to a centralized ADC is shown in table 2, the state of the art using power line communication (PLC) [1, 2] is included in table 3 and the materials required with

the proposed talkative battery are listed in table 4. The battery pack contains 12 of the cells used in the paper, each of which has 4 temperature sensors attached. A battery management system for 12 to 24 cells with a maximum current of 100A suitable for LFP cells is considered.

Table 1: Main components of the battery pack, which are not related to temperature measurement.

Component	Source/Digikey-ID	Quantity	Single Cost	Total Cost
CA100AHA Battery Cell	[A]	12	188.78 €	2.265€
Cable Main 50 mm ²	[B]	3m	9.5 €/m	28.5 €
Cable Balancing 6mm ²	[C]	5m	0.85 €/m	4.25 €
Crimp-on ring terminal	281-1493150000-ND	26	1.65 €	36.30 €
BMS LiFePO 12-24S 100A	[D]	1	255 €	255 €
Housing increment		1	100 €	100 €
Total				2.589.41€

Table 2: Additional components required for temperature measurement following the conventional method.

Component	Source/Digikey-ID	Quantity	Single Cost	Total Cost
Thermistor NTC	317-1258-ND	48	0.19 €	9.12 €
Thermistor Cables	135-XCZ1E031000-ND	48 · 0.6m	0,52€/m	14.97 €
12 Channel ADC	296-25852-2-ND	4	4.70 €	18.08 €
T-Supervision MCU	ATMEGA328PB-MU-ND	1	1.3 €	1.3 €
Total				43.47€

Table 3: Additional components required for temperature measurement following the state of the art. Materials are selected according to the suggested design in [2]. Replacing the ADC and MCU with the ATMEGA328P MCU used in the proposed solution might reduce costs, however, the PLC modems keep the price of this methodology high.

Component	Source/Digikey-ID	Quantity	Single Cost	Total Cost
Thermistor NTC	317-1258-ND	48	0.19 €	9.12 €
PLC Modem	296-THVD8000DDFRTR-ND	13	3.3€	42.9€
4 Channel ADC	TLA2024IRUGR	12	1.3 €	15.6 €
MCU	ATSAMD21E18A-MUT	12	3 €	36 €
Total				103.62€

Table 4: Additional components required using the proposed temperature data transmission approach.

Component	Source/Digikey-ID	Quantity	Single Cost	Total Cost
THERMISTOR NTC	317-1258-ND	48	0.19 €	9.12 €
Talkative battery MCU	ATMEGA328PB-MU-ND	12	1.3 €	15.6 €
MOSFET	3141-G5N02LCT-ND	12	0.128 €	1.536 €
Capacitor	GRM155R61A564ME15D	12	0.04 €	0.48€
Inductor cable	[E]	12 · 0.4m	0.27€/m	1.296 €
Total				28.03€

In conclusion, the talkative battery realization adds only 28 €, while conventional solution adds 43€ and the solution based on PLC adds 102€. These results indicate that the internal sensing can be achieved with $\frac{2}{3}$ of the cost of the conventional solutions and only 27% of the state of the art solution. Although the costs of the sensors are generally much smaller than the battery pack itself, in large scale production, it could lead to tangible change in the overall costs and competitiveness.

Component Sources

- [A] <https://www.evea-solutions.com/en/lithium-cells/1485-calb-lithium-cell-ca-3-2v-100ah-lifepo4.html>
- [B] https://batterie24.de/kabel-h07v-k-feindrachtig-50mm2-rot/90418?gad_source=1&gad_campaignid=23510943485&gclid=CjwKCAjw1N7NBhAoEiwAcPchpytn7v08_2gb2CeqlSrpD9DEvMtgPIZD3wR0d8XcK8KxVTvu152DrBoCfrMQAvD_BwE
- [C] <https://www.digikey.de/de/products/detail/leader-solar-cable-and-connector/FGUL005/16179575?s=N4IgjCBco0wBxVAYygMwIYBsDOBTANCAPZQDaIMazAAzwQC6hADgC5QgDKLATgJYB2AcxABfEYQBMZEEEnQAjTLhCMQAVkQgmUMMy2QJqsUA>
- [D] <https://www.i-tecc.de/shop/bmspcm/bms-lifepo4/12s-36v/413/bms-lifepo-12-24s-100a-36-72v>
- [E] https://www.reichelt.de/de/de/shop/produkt/kupferlackdraht_0_7mm_laenge_12m-9619?PR0VID=2788&gad_source=1&gad_campaignid=18347782207&gclid=CjwKCAjw687NBhB4EiwAQ645dnYgStZBEPPoZ537HmRdUpXrYehMibtzxz9i5sk06BVh-T3_L6RsTuhoc-mIQAvD_BwE

References

1. Landinger, T. F. *et al.* *Power Line Communications in Automotive Traction Batteries: A Proof of Concept* in *2020 IEEE International Symposium on Power Line Communications and its Applications (ISPLC) 2020 IEEE International Symposium on Power Line Communications and its Applications (ISPLC)* (IEEE, Malaga, Spain, May 2020), 1–5. ISBN: 978-1-72814-816-8. <https://ieeexplore.ieee.org/document/9115412/> (2024).
2. Vincent, T. A., Gulsoy, B., Sansom, J. E. H. & Marco, J. A Smart Cell Monitoring System Based on Power Line Communication-Optimization of Instrumentation and Acquisition for Smart Battery Management. *IEEE Access* **9**, 161773–161793. ISSN: 2169-3536. <https://ieeexplore.ieee.org/document/9627991/> (2024) (2021).